# DiScan: Quad-directional SSM Diffusion with Retrieval-Augmented Cross-Scanning for Efficient Text-to-Image Synthesis

## Abstract

Text-to-image synthesis models often suffer from texture blurring, shape distortion, and poor alignment with textual prompts. These issues stem from limited spatial modeling, weak cross-modal interaction, and insufficient detail preservation. To address them, we propose DiScan, a framework combining directional state-space modeling with retrieval-based fusion for efficient, high-fidelity synthesis. First, we introduce a quad-directional SSM that jointly scans visual and textual features across directions. It shares dynamics for parameter efficiency and uses direction-specific projections to enhance spatial coherence and semantic consistency. Second, we design a dual-stage attention module using retrieved references. The first stage aligns prompt and image features via cross-attention. The second modulates features through direction-aware scanning, improving structure preservation. Third, we propose a spatial-frequency fusion block that combines wavelet decomposition with bidirectional scanning. It captures fine textures and enhances local details. Extensive experiments show DiScan outperforms Zigma and USM, achieving significant FID improvements (+6.3 on CelebA-HQ, +9.95 on COCO, +0.67 on CIFAR-10) while maintaining excellent visual quality. Our work establishes directional SSM diffusion as a scalable paradigm for efficient high-fidelity synthesis.

## 1 Introduction

Diffusion models have significantly advanced text-to-image synthesis. These models can now synthesize highly realistic images from complex prompts Saharia et al. (2022); Rombach et al. (2022). Most approaches follow the latent diffusion framework, which performs the diffusion process in a compressed latent space. This design reduces computational costs during both training and inference. To improve scalability, recent works have replaced the traditional U-Net backbone with transformer-based architectures Peebles & Xie (2023). Transformers enable more flexible modeling and larger receptive fields. However, self-attention has quadratic complexity with respect to input size. This limitation becomes critical when synthesizing high-resolution images Shen et al. (2021). As a result, balancing synthesis quality with efficiency remains an open challenge.

Recently, State Space Models (SSMs) Gu et al. (2022; 2021) have gained attention for their efficiency in long-sequence modeling. Mamba Gu & Dao (2023) is a notable example. It achieves remarkable transformer-level performance while using significantly fewer resources. Its linear-time complexity allows lower memory consumption compared to traditional attention. Although originally designed for 1D sequences, Mamba has been extended to 2D tasks and shows strong results in classification, detection Liu et al. (2024).

Building on the efficiency of Mamba, recent works have explored its application to image synthesis. Zigma Hu et al. (2024) integrates Mamba into diffusion models for synthesis. It adopts an efficient row-column scanning pattern to reduce computation. However, this unidirectional scanning breaks spatial continuity. It leads to inconsistent modeling of local structures such as object boundaries. The issue stems from fixed tokenization paths Doruk & Ates (2025). Multi-directional variants Fei et al. (2024b) attempt to address this, but their independent scanning limits interaction between features. Furthermore, due to simple cross-attention mechanisms, these models struggle to align

textual descriptions with image content. This results in object omission and attribute mismatches, especially in complex prompts Ergasti et al. (2025), as shown in Figure 1(a).

Following the trend of lightweight synthetic architectures, SSM-based image synthesis models adopt spatial feature scanning over visual tokens to balance quality and efficiency Fei et al. (2024a), Teng et al. (2024), Li et al. (2024). Compared to traditional DiT Peebles & Xie (2023) and U-ViT Bao et al. (2023), they provide higher image fidelity and improved computational efficiency. These models achieve competitive results with lower memory consumption. However, their reliance on fixed spatial scanning still limits their ability to capture fine-grained structures. In particular, detailed features such as textures and edges are often blurred or missing. This leads to a drop in visual quality in local regions, especially those with rich textures or detail structural variations, as shown in Figure 1(b).

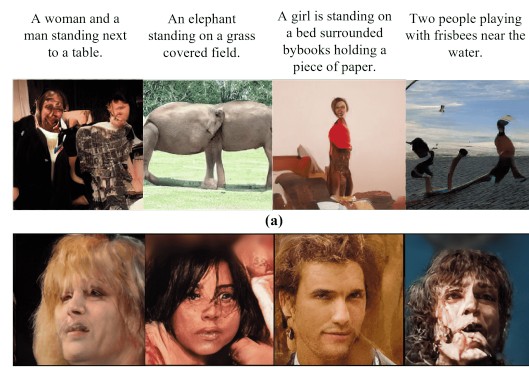

Figure 1: Failure cases of text-to-image synthesis models on MS-COCO and MM-CelebA datasets. **(a)** Synthesized images show broken structures and unrealistic body deformations. This includes distorted limbs, misaligned faces, and fused objects. **(b)** Facial synthesis results contain blurred features, asymmetry, and unnatural skin textures. These issues reflect poor alignment between the text prompt and generated content. High-frequency features in textures are also poorly handled.

To tackle the aforementioned issues, we present DiScan, a novel text-to-image synthesis framework with three targeted innovations. First, we design a Shared-Projection Directional State-Space **(SP-DSS)** Fusion mechanism to improve spatial continuity and visual-text alignment. This component scans visual features along four orthogonal directions. During this process, it jointly integrates text embeddings with visual tokens. We share parameters across directional pathways to reduce redundancy and improve training efficiency. At the same time, we use direction-specific projection layers. These layers encode unique spatial information and maintain feature diversity. As a result, the model gains a more complete understanding of object shapes, textures, and orientations. Second, we introduce a Dual-Phase Retrieval-Augmented Conditioning **(DP-RAG)** module to address object deformation and poor alignment. This module retrieves structural cues from reference data, such as object layout or pose. It then modulates intermediate features in two stages. The first stage performs semantic alignment between the text prompt and the retrieved structure. The second stage applies geometric conditioning to guide spatial arrangement. By decoupling structure injection into two steps, we avoid introducing noise and preserve the integrity of fine object structures. Third, we incorporate a Unified Spectral-Spatial Co-Processing **(USS-CP)** block to capture high-frequency details. These blocks operate at selected layers of the network. They first perform frequency decomposition to extract texture-related signals. Then, spatial analysis modules refine visual patterns such as edges and contours. A unified state-transition operation combines both types of information. This enhances the model's ability to retain detailed textures, which are often blurred or lost in traditional methods. Together, these three components enable DiScan to produce high-quality, semantically aligned images with realistic structures and textures.

In summary, Our principal contributions are:

- We propose a novel multi-directional fusion architecture that jointly scans image and text features through parameter-shared state-space dynamics, effectively reducing redundancy while enhancing cross-modal alignment and spatial consistency in synthesized images.

- We design a dual-phase retrieval conditioning framework that first aligns semantic content and then injects structural cues, preserving object shapes and layouts without introducing artifacts from direct feature fusion.

- We introduce an integrated spectral-spatial enhancement module that combines wavelet-based frequency decomposition with directional scanning, addressing the loss of texture and fine details in high-frequency regions.

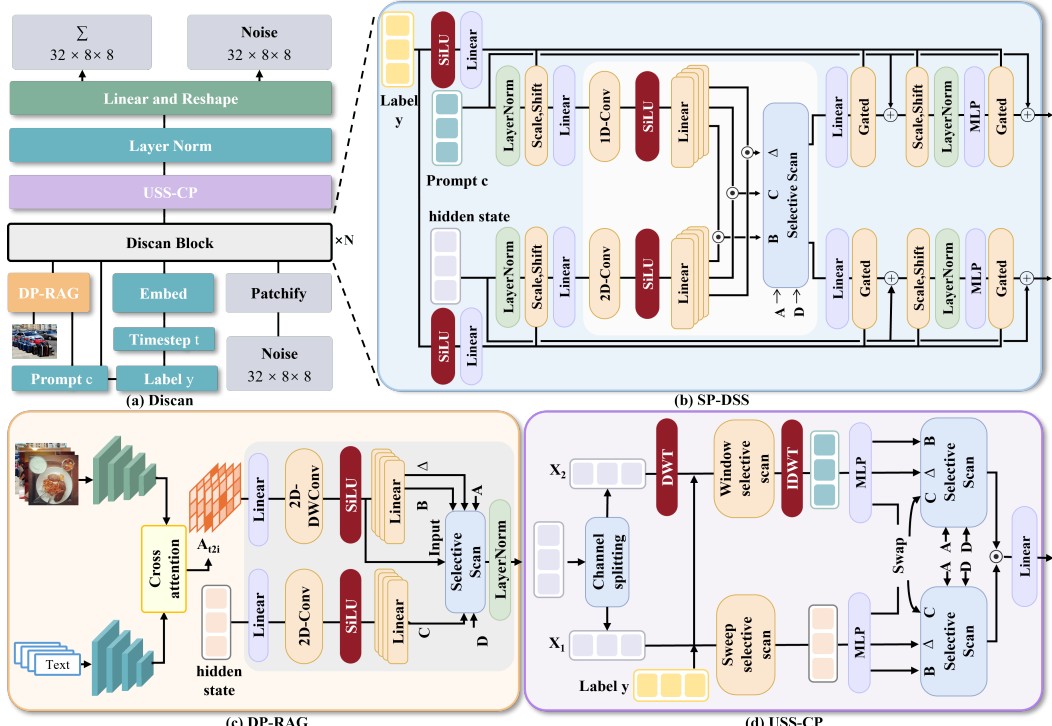

Figure 2: (a) Overview of our proposed Discan framework. (b) One Discan block. (c) Overall architecture of the DP-RAG module. (d) Overall architecture of the USS-CP module.

- We achieve state-of-the-art synthesis performance among SSM-based methods, with notable gains in both quality and efficiency, including +9.95 FID on COCO and +6.3 FID on CelebA-HQ.

## 2 RELATED WORK

### 2.1 STATE SPACE MODEL (SSM).

Transformers have shown strong performance in vision tasks Dosovitskiy et al. (2021), but their quadratic complexity limits efficiency. State Space Models (SSMs) Gu et al. (2022; 2021) offer linear-time computation. Mamba Gu & Dao (2023) improves SSMs with input-dependent dynamics and achieves high performance in tasks like classification, detection, and segmentation Botti et al. (2025); Guo et al. (2024); Liu et al. (2024). Zigma Hu et al. (2024) and Dimba Fei et al. (2024b) extend Mamba to diffusion, reducing computation cost. However, existing variants often suffer from semantic misalignment and structure distortion. We address this with multi-directional fusion and structure-aware retrieval to enhance spatial consistency and object accuracy.

### 2.2 SPATIAL-FREQUENCY METHODS.

SSM-based text-to-image models scan spatial tokens to improve quality and lower cost Fei et al. (2024a); Teng et al. (2024); Li et al. (2024). They outperform transformer baselines in memory and fidelity. However, spatial-only scanning fails to capture fine textures and sharp edges. Hybrid methods from other domains show that combining spatial and frequency features improves detail recovery and efficiency Liu et al. (2022; 2023); Yao et al. (2022). Recent works integrate frequency into SSMs using Fourier or wavelet techniques Patro & Agneeswaran (2024); Zou et al. (2024). Inspired by this, we design a spectral-spatial fusion module for diffusion. It improves texture quality and object boundaries by combining frequency decomposition and spatial modeling.

## 3 PRELIMINARIES

### 3.1 STATE SPACE MODEL (SSM).

State Space Models (SSMs) model sequences by simulating continuous systems through linear ODEs. For an input $x(t) \in \mathbb{R}$, SSMs maintain a latent state $h(t) \in \mathbb{R}^N$ evolving over time, producing output $y(t) \in \mathbb{R}$. They achieve linear complexity via recurrent state propagation, unlike transformers' quadratic scaling. Continuous dynamics are:

$$h'(t) = \mathbf{A}h(t) + \mathbf{B}x(t), \quad y(t) = \mathbf{C}h(t) + \mathbf{D}x(t), \tag{1}$$

where $\mathbf{A}, \mathbf{B}, \mathbf{C}, \mathbf{D}$ are learnable parameters. These equations are typically discretized using zero-order hold (ZOH), allowing efficient step-wise recurrence. To enhance input adaptability, Mamba Gu & Dao (2023) introduces input-conditioned dynamics by making key parameters depend on the current input:

$$\mathbf{B}_k = \mathbf{W}_B x_k, \quad \mathbf{C}_k = \mathbf{W}_C x_k, \quad \Delta_k = \mathbf{W}_\Delta x_k, \tag{2}$$

where $\mathbf{W}_B, \mathbf{W}_C, \mathbf{W}_\Delta$ are learnable projection matrices. This design enables Mamba to selectively modulate transitions based on content, improving context awareness while maintaining linear-time complexity. It is particularly effective for modeling long-range dependencies in high-resolution synthesis tasks.

### 3.2 WAVELET TRANSFORMATION.

Wavelet transforms (WT) excel in simplicity and efficiency among frequency transformation methods. By preserving spatial structure, WT decomposes images into low-frequency approximations ($LL$) and high-frequency detail components ($LH, HL, HH$) capturing vertical, horizontal, and diagonal edges. Haar wavelets, the most common WT variant, apply basic low-pass and high-pass filtering to extract different frequency bands. These operations divide an image into four subbands, which represent coarse structure and fine details from multiple directions. The transformation is fully reversible, allowing exact reconstruction through the inverse wavelet process.

## 4 PROPOSED METHOD

### 4.1 OVERALL ARCHITECTURE

We propose DiScan, a novel text-to-image synthesis framework. It leverages state-space models for efficient long-range dependency modeling and introduces directional scanning to improve cross-modal synthesis. As illustrated in Figure 2(a), our architecture integrates three core innovations: (i) a multi-directional state-space fusion backbone for joint text-visual representation learning, (ii) a retrieval-augmented geometric conditioning module for structural consistency, and (iii) a spectral-spatial co-processing unit for high-frequency texture refinement.

Given an input text prompt $T$, the textual embedding is obtained via linear projection $\mathbf{E}_t = W_t(T)$, where $W_t$ is a learnable projection matrix. For visual input, we use DC-AE Chen et al. (2025) to extract latent features efficiently. The input image $I \in \mathbb{R}^{H \times W \times 3}$ is first projected, then encoded as:

$$\mathbf{E}_v = \text{DC-AE}(W_v(I)) \in \mathbb{R}^{\frac{H}{32} \times \frac{W}{32} \times d}, \tag{3}$$

where $W_v$ denotes a learnable linear projection that maps pixel values to a $d$-dimensional space.

### 4.2 SP-DSS FUSION

**Shared Projection for Cross-Modal Alignment.** Unlike Zigma's convolution-only design Hu et al. (2024), which overlooks text semantics, our parameter-shared state-space framework achieves efficient cross-modal alignment in Figure 2(b). Given text embeddings $\mathbf{E}_t \in \mathbb{R}^{L_t \times d}$ and visual embeddings $\mathbf{E}_v \in \mathbb{R}^{\frac{H}{32} \times \frac{W}{32} \times d}$, we refine each modality as:

$$\mathbf{Z}_t = \text{SiLU}(\text{Conv}_{1d}(\text{Linear}_t(\mathbf{E}_t))), \mathbf{Z}_v = \text{SiLU}(\text{Conv}_{2d}(\text{Linear}_v(\mathbf{E}_v))), \tag{4}$$

where $\text{Conv}_{1d}$ and $\text{Conv}_{2d}$ process text and vision features separately. This retains modality structure, which Zigma lacks during fusion. The parameterization comprises:

**(1)Shared Dynamics Parameters**: For each scanning direction $i \in \{1, 2, 3, 4\}$, system matrices $\mathbf{A}^{(i)} \in \mathbb{R}^{n \times n}$ and discretization factors $\Delta^{(i)} \in \mathbb{R}^n$ are defined with sequence length $L = L_t + L_v$ modeling cross-modal state transitions:

$$\mathbf{A}^{(i)}, \Delta^{(i)} \quad \text{for} \quad i = 1, 2, 3, 4. \tag{5}$$

These parameters are universally shared across both textual and visual modalities.

**(2) Direction-Specific Projections**: For each direction $i \in \{1, 2, 3, 4\}$, we compute modality-specific parameters from refined features $\mathbf{Z}_t$ and $\mathbf{Z}_v$:

$$\left[\mathbf{B}_t^{(i)} \ \mathbf{C}_t^{(i)} \ \mathbf{X}_t^{(i)}\right] = W_t^{(i)}(\mathbf{Z}_t), \left[\mathbf{B}_v^{(i)} \ \mathbf{C}_v^{(i)} \ \mathbf{X}_v^{(i)}\right] = W_v^{(i)}(\mathbf{Z}_v), \tag{6}$$

where $W_t^{(i)}$ and $W_v^{(i)}$ are learnable direction-specific projections for text and vision, respectively. The final directional parameters are concatenated across modalities:

$$[\mathbf{B}^{(i)}, \mathbf{C}^{(i)}, \mathbf{X}^{(i)}] = \text{concat}([\mathbf{B}_t^{(i)}, \mathbf{B}_v^{(i)}], [\mathbf{C}_t^{(i)}, \mathbf{C}_v^{(i)}], [\mathbf{X}_t^{(i)}, \mathbf{X}_v^{(i)}]), \tag{7}$$

where the fused parameters are processed by the SSM layer, followed by layer normalization and output projection. The output is split into text/visual components $\mathbf{O}_t, \mathbf{O}_v$, each enhanced through gated modulation:

$$\mathbf{O}_k' = \sigma(\gamma_k) \odot \mathbf{O}_k + \beta_k, \quad k \in \{t, v\}, \tag{8}$$

where $\gamma_k, \beta_k$ are modality-specific affine parameters, establishing bidirectional grounding-text constrains visual structure synthesis while visual context refines language understanding-with only $\mathcal{O}(L)$ complexity.

**Bias-Free Multi-Directional Feature Fusion Mechanism.** Conventional unidirectional scanning imposes spatial inductive bias by privileging specific traversal orders. To achieve isotropic feature integration, we design a quadrangular scanning protocol that processes sequences along four orthogonal directions as shown in Figures 3. For each direction $i \in \{1, 2, 3, 4\}$, the transformation and fusion are computed as:

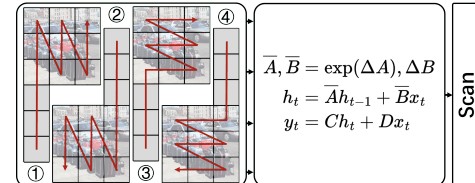

$$\mathbf{Y}^{(i)} = \text{SSM}_{\theta_i}(\mathbf{P}^{(i)}), \quad \mathbf{Y} = \sum_{i=1}^{4} \mathcal{R}_i^{-1}(\mathbf{Y}^{(i)}), \tag{9}$$

Figure 3: The 2D-selective scan with a 3×3 example image.

where $\theta_i = \{\mathbf{A}, \Delta, \mathbf{B}^{(i)}, \mathbf{C}^{(i)}, \mathbf{X}^{(i)}\}$ contains direction-specific parameters (with $\mathbf{A}$ and $\Delta$ shared), and $\mathcal{R}_i^{-1}$ are inverse permutation operators restoring spatial coordinates. This symmetric fusion eliminates directional priors while preserving spatial coherence through bidirectional traversals, enhancing cross-modal alignment.

## 4.3 DP-RAG MODULE

**Semantic Priming via Cross-Attention.** To align text with references, we apply cross-attention priming as shown in Figure 2(c). The input text embedding $\mathbf{E}_t \in \mathbb{R}^{L_t \times d}$ and the reference feature $\mathbf{F}_{\text{ref}} \in \mathbb{R}^{H_r \times W_r \times c}$ are used. We project $\mathbf{E}_t$ to queries and flatten $\mathbf{F}_{\text{ref}}$ into a sequence to compute keys and values:

$$\mathbf{Q} = \mathbf{W}_q(\mathbf{E}_t), \quad \bar{\mathbf{F}}_{\text{ref}} = \text{Flatten}(\mathbf{F}_{\text{ref}}), \quad \mathbf{K} = \mathbf{W}_k(\bar{\mathbf{F}}_{\text{ref}}), \quad \mathbf{V} = \mathbf{W}_v(\bar{\mathbf{F}}_{\text{ref}}), \tag{10}$$

We then compute $\mathbf{A}_{\text{prim}}$ via standard scaled dot-product attention between $\mathbf{Q}$, $\mathbf{K}$, and $\mathbf{V}$, which builds semantic links between text and visual features, serving as guidance for later spatial alignment.

**Structural Propagation via Cross-Mamba Modulation.** Building on semantic anchors $\mathbf{A}_{\text{prim}} \in \mathbb{R}^{L_t \times d}$, we apply cross-mamba to guide visual features $\mathbf{E}_v \in \mathbb{R}^{h \times w \times d}$ with geometric priors. Visual content is projected as $\mathbf{C} = W_c(\mathbf{E}_v)$, while $\mathbf{A}_{\text{prim}}$ is transformed as:

$$\mathbf{F}_{\text{down}} = \text{Conv}_{2d}(\mathbf{A}_{\text{prim}}), \mathbf{B} = W_b(\mathbf{F}_{\text{down}}), \mathbf{X} = W_x(\mathbf{F}_{\text{down}}), \tag{11}$$

where $\mathbf{C}$ encodes spatial content, and $\mathbf{B}, \mathbf{X}$ provide structure-aware modulation. These parameters are subsequently used in the SSM module to integrate geometric structure. This alleviates object deformation and misalignment by incorporating structural priors from reference data through decoupled semantic and geometric conditioning.

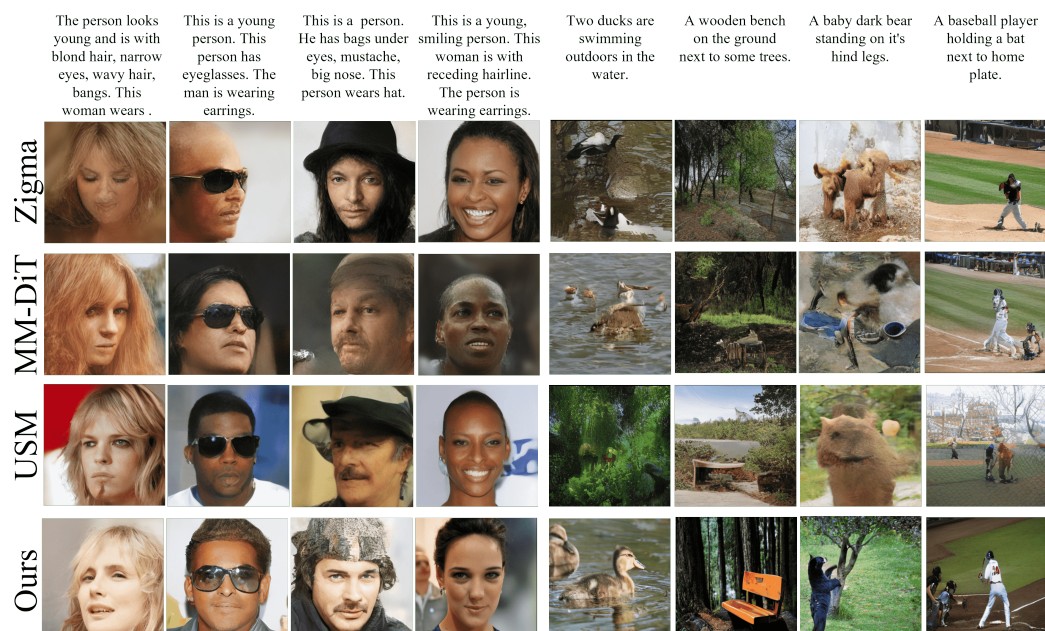

Figure 4: Qualitative comparison on the MM-CeleBA and MS-COCO dataset. The input text descriptions are given in the first row and the corresponding synthesized images from different methods are shown in the same column.

## 4.4 USS-CP BLOCK

**Frequency-Space Decomposition with Directional DWT.** To resolve texture-structure trade-offs, directional DWT decomposition processes intermediate features $\mathbf{F} \in \mathbb{R}^{C \times H \times W}$ via Haar wavelets in Figure 2(d):

$$\{\mathbf{F}_{\text{LL}}, \mathbf{F}_{\text{LH}}, \mathbf{F}_{\text{HL}}, \mathbf{F}_{\text{HH}}\} = \text{DWT}(\mathbf{F}), \tag{12}$$

where $\mathbf{F}_{\text{LL}}$ captures low-frequency structural contours, and $\mathbf{F}_{\text{LH}}, \mathbf{F}_{\text{HL}}, \mathbf{F}_{\text{HH}}$ encode high-frequency textures along horizontal/vertical/diagonal axes. This separates global shape priors from local textures, overcoming spatial-only scanning limitations. Subbands are processed via parallel directional scanning for frequency-specific texture synthesis.

**Cross-Mamba Enhanced Texture Synthesis.** To fuse spatial and spectral features, a dual-path cross-mamba architecture processes spatial features $\mathbf{F}_{\text{sp}}$ and wavelet subbands $\mathbf{F}_{\text{freq}}$. Two state-space projections are computed with shared dynamics but cross-modality parameters:

$$\begin{aligned}
\mathbf{C}_{\text{sp}} &= W_c(\mathbf{F}_{\text{sp}}), \quad [\mathbf{B}_{\text{freq}}, \mathbf{X}_{\text{freq}}] = W_{bx}(\mathbf{F}_{\text{freq}}), \\
\mathbf{Y}_a &= \text{SSM}_{\Theta_a}(\mathbf{X}_{\text{freq}}), \quad \Theta_a = \{\mathbf{A}, \Delta, \mathbf{B}_{\text{freq}}, \mathbf{C}_{\text{sp}}\}, \\
\mathbf{Y}_b &= \text{SSM}_{\Theta_b}(\mathbf{X}_{\text{sp}}), \quad \Theta_b = \{\mathbf{A}, \Delta, \mathbf{B}_{\text{sp}}, \mathbf{C}_{\text{freq}}\}, \\
\mathbf{Y} &= W_{\text{fuse}}([\mathbf{Y}_a; \mathbf{Y}_b]),
\end{aligned} \tag{13}$$

where $\mathbf{C}_{\text{freq}}, \mathbf{B}_{\text{sp}}, \mathbf{X}_{\text{sp}}$ are similarly obtained via linear projections. Cross-parameterization allows spatial features to guide frequency synthesis, while spectral cues regularize spatial structure. This enhances the preservation of fine textures and boundaries by jointly modeling spatial layouts and high-frequency signals, mitigating the blurring effects seen in conventional generation pipelines.

## 4.5 INFERENCE STRATEGY.

DiScan employs dual retrieval pathways: one retrieves the most semantically aligned reference image via CLIP text-image similarity; the other fetches the top text description through CLIP text-text similarity and its paired image. Both use top-1 retrieval to augment structural conditioning via the dual-phase module.

Table 1: Performance comparison on MM-CelebA datasets. Our method can outperform the baseline and can achieve even better results

| Methods | MultiModal-CelebA-256 | | |
|---|---|---|---|
| | $\text{FID}^{5k} \downarrow$ | $\text{FDD}^{5k} \downarrow$ | $\text{KID}^{5k} \downarrow$ |
| Sweep | 158.1 | 75.9 | 0.1690 |
| Zigzag-1 | 65.7 | 47.8 | 0.0510 |
| Zigzag-2 | 54.7 | 45.5 | 0.0410 |
| Zigma | 45.5 | 26.4 | 0.0110 |
| USM | 13.6 | 17.3 | 0.0051 |
| MM-DiT | 16.6 | 20.3 | 0.0088 |
| $\text{DiScan}_{t2i}$ | 22.1 | 16.8 | 0.0040 |
| **$\text{DiScan}_{t2t}$** | **7.3** | **4.7** | **0.0022** |

Table 2: Performance comparison on MS-COCO datasets. Our method consistently outperforms the baseline.

| Methods | MS-COCO2014 | |
|---|---|---|
| | Images | $\text{FID}^{5k} \downarrow$ |
| Sweep | $400\text{K} \times 256$ | 195.10 |
| Zigzag-1 | $400\text{K} \times 256$ | 73.10 |
| VisionMamba | $400\text{K} \times 256$ | 60.20 |
| Zigma | $400\text{K} \times 256$ | 41.80 |
| USM | $400\text{K} \times 8$ | 39.10 |
| MM-DiT | $150\text{K} \times 256$ | 29.26 |
| $\text{DiScan}_{t2i}$ | $150\text{K} \times 256$ | 25.37 |
| **$\text{DiScan}_{t2t}$** | $150\text{K} \times 256$ | **19.31** |

## 5 EXPERIMENTS

### 5.1 EXPERIMENTAL SETUP

**Datasets.** We conduct comprehensive training and evaluation on three benchmark datasets widely adopted in text-to-image synthesis research:

- **MM-CelebA-HQ** Xia et al. (2021) extends the CelebA-HQ dataset with multimodal annotations, containing 30,000 high-fidelity celebrity facial images. Each image is paired with semantic-rich textual descriptions, providing granular control for attribute-driven facial generation.

- **MS-COCO2014** Lin et al. (2014) leverages the multimodal subset of the COCO dataset, comprising approximately 330,000 diverse natural scene images. Crucially, each image includes 5 human-annotated captions detailing object interactions and contextual semantics, enabling robust evaluation of complex text-to-scene synthesis.

- **CIFAR-10** Krizhevsky et al. (2009) serves as a stress-test benchmark with 50,000 low-resolution images ($32 \times 32$) across 10 object categories. We synthesize minimalistic textual prompts (e.g., "a photo of a [class]") to evaluate model performance under constrained visual-textual alignment scenarios.

**Implementation Details.** To ensure architectural consistency with contemporary models, our framework adopts identical core configurations: a 24-layer deep architecture with 768-dimensional embedding spaces. All experiments employ the AdamW optimizer with a fixed learning rate of $1 \times 10^{-4}$ for parameter updates. For MM-CelebA and MS-COCO datasets, models were trained for 150K iterations at $256 \times 256$ resolution, while CIFAR-10 training of 500K iterations. Image synthesis utilizes stochastic differential equations (SDE) with 25 sampling steps during inference.

**Evaluation Metrics.** We employ three complementary metrics: FID Heusel et al. (2017) and KID Bińkowski et al. (2018) (measuring distributional similarity on 5k real/synthetic images), supplemented by FDD Hu et al. (2024) to better address FID's perceptual limitations.

Table 3: Performance comparison on CIFAR-10 datasets.

| Methods | CIFAR-10 | |
|---|---|---|
| | Type | $\text{FID}^{5w} \downarrow$ |
| DDPM | U-Net | 3.27 |
| EDM | U-Net | 2.10 |
| GenViT | Transformer | 20.20 |
| U-ViT | Transformer | 2.87 |
| DiM | SSM | 2.76 |
| DiS | SSM | 3.17 |
| **Ours** | **SSM** | **2.09** |

FDD leverages DINOv2 features Oquab et al. (2024) for enhanced semantic and structural alignment with human evaluation.

**Baseline Methods.** Our approach is compared with state-of-the-art models including: ZigMa Hu et al. (2024) (DiT-style Zigzag diffusion) and its variants Zigzag-1/2; Sweep-based methods; Vision-

Mamba Zhu et al. (2024) (vision state-space model); USM Ergasti et al. (2025) (U-shape diffusion); MM-DiT Esser et al. (2024) (multimodal Rectified Flow diffusion); classical diffusion models DiS Fei et al. (2024a) (state-space backbone) and DiM Teng et al. (2024) (Diffusion Mamba for HR image synthesis); and transformer architectures U-ViT-S/2 Bao et al. (2023), GenViT Yang et al. (2022) and DDPM Ho et al. (2020).

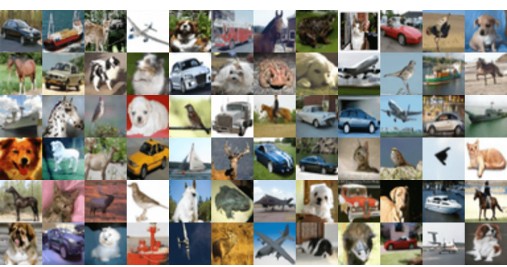

Figure 5: Image synthesis results of DiScan on CIFAR-10.

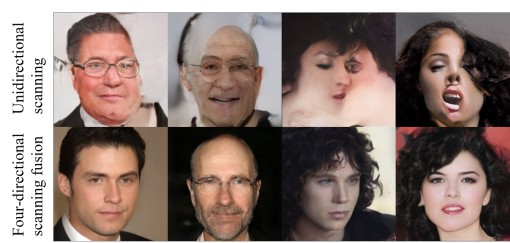

Figure 6: Visualization of the comparison between single-direction scanning (top) and four-directional fusion (bottom).

## 5.2 COMPARISONS

**Qualitative Results.** We compare our method with state-of-the-art (SOTA) baseline methods, as shown in Figures 4 and 5. Visual comparisons reveal critical limitations in baseline methods: Zigma exhibits inconsistent text alignment and structural distortions-synthesized faces partially match descriptions but objects display chaotic arrangements (1st row). MM-DiT suffers from facial blurring artifacts and impoverished diversity, with rigid compositions failing to preserve object integrity (2nd row). While USM enhances visual richness, it compromises semantic fidelity—objects deviate from textual guidance (3rd row, 8th col). In contrast, DiScan maintains precise attribute control for facial

Table 4: Comparison of Semantic Fidelity (CLIPScore) and Perceptual Quality (NIQE)

| Methods | MultiModal-CelebA-256 | |
| --- | --- | --- |
| | CLIPScore$^{5k}$ ↑ | NIQE$^{5k}$ ↓ |
| Zigma | 21.02 | 8.33 |
| MM-DiT | 22.24 | 6.31 |
| **Ours** | **22.43** | **6.18** |

features , enriches scenes with dynamic elements , and preserves structural coherence across entities. Notably on CIFAR-10, our model accurately synthesizes small-scale features.

**Quantitative Results.** DiScan demonstrates superior performance across all benchmarks, achieving state-of-the-art metrics on MM-CelebA, MS-COCO, and CIFAR-10, as shown in Table 1, 2 and 3. On MM-CelebA, our $DiScan_{t2t}$ variant reduces FID to 7.3, improves FDD to 4.7, and lowers KID to 0.0022. For MS-COCO, $DiScan_{t2t}$ achieves FID 19.31 at equivalent computational cost, outperforming MM-DiT and USM. On CIFAR-10, DiScan sets a new record with FID 2.09, surpassing all SSM-based competitors including DiM and transformer-based U-ViT-S/2, validating its efficiency in low-resolution domains. The results demonstrate the performance advantage of our proposed method.

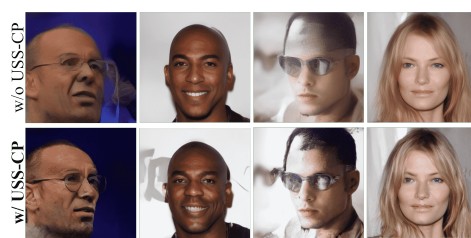

Figure 7: Visual Ablation Study for the Proposed USS-CP Method.

**Analysis of Semantic Fidelity and Texture Detail.** We present quantitative results on MultiModal-CelebA-256 in Table 4, comparing our method against Zigma and MM-DiT. Our approach achieves the highest CLIPScore, reflecting improved cross-modal alignment through our multi-directional fusion architecture. Simultaneously, the best NIQE score demonstrates superior perceptual quality, attributed to our spectral-spatial enhancement module that preserves fine details. These results validate the effectiveness of our innovations in jointly optimizing semantic fidelity and visual realism.

**Analysis of Directional Fusion Comparisons.** We compare single-direction scanning with our proposed four-directional fusion. As shown in Fig. 6, single-direction results suffer from noticeable distortions and inconsistent structures, while multi-directional fusion consistently yields sharper faces, better-preserved shapes, and improved spatial alignment. This clearly demonstrates the effectiveness of directional fusion in enhancing visual fidelity.

## 5.3 ABLATION STUDY

## 5.4 COMPARISONS

**Effectiveness of SP-DSS.** The integration of our shared-projection directional state-space fusion (SP-DSS) significantly enhances cross-modal alignment, as clearly shown by MM-CelebA. When replacing Zigma's convolutional scanning and cross-attention with SP-DSS and training for the same steps, FID improves from 84.4 to 15.7, FDD drops from 60.6 to 24.7, and KID decreases from 0.0768 to 0.0074. As shown in Table 5 and Figure 8, removing this module disrupts text-image semantic coherence, further validating the critical role in synchronizing linguistic concepts with visual structure through parameter-shared directional scanning.

Table 5: Ablation studies on architecture at Multi-Modal-CelebA-HQ datasets.

| Methods | MultiModal-CelebA-256 | | |
|---|---|---|---|
| | FID$^{5k}$ ↓ | FDD$^{5k}$ ↓ | KID$^{5k}$ ↓ |
| Baseline | 84.4 | 60.6 | 0.0768 |
| + SP-DSS | **15.7** | 24.7 | 0.0074 |
| + DP-RAG | 22.7 | 16.9 | 0.0048 |
| **+ USS-CP** | 22.1 | **16.8** | **0.0040** |

**Effectiveness of DP-RAG.** The dual-phase retrieval-augmented conditioning (DP-RAG) module critically enhances structural fidelity, as strongly evidenced by its impact on MM-CelebA metrics. Adding DP-RAG to the SP-DSS foundation reduces FDD by 31% and KID by 35% , though FID increases correspondingly from 15.7 to 22.7 due to heightened structural constraints limiting sample diversity. As shown in Table 5, This further confirms DP-RAG's efficacy in anchoring subject consistency without direct feature injection bottlenecks.

Figure 8: Visual ablation study for the SP-DSS.

**Effectiveness of USS-CP.** The unified spectral-spatial co-processing (USS-CP) module significantly enhances high-frequency texture fidelity, as demonstrated by its impact. Adding USS-CP to the DP-RAG foundation improves KID by 17% while maintaining optimal FDD (16.8) and slightly reducing FID. As shown in Table 5 and Figure 7, removing USS-CP causes critical texture degradation—due to inadequate high-frequency component integration. This validates USS-CP's role in harmonizing wavelet-domain texture synthesis with spatial structure through unified state transitions, eliminating artifacts while preserving stochastic details.

## 6 CONCLUSIONS

We present DiScan, a state-space model-based text-to-image framework redefining cross-modal synthesis via three innovations: 1) Multi-directional fusion harmonizes text-visual semantics using parameter-shared quadrangular scanning to eliminate spatial bias; 2) Dual-phase retrieval conditioning ensures geometric consistency through hierarchical semantic-to-geometric modulation; 3) Integrated spectral-spatial enhancement elevates texture fidelity via unified frequency-space co-processing, resolving structure-detail trade-offs. This establishes new fidelity standards, enabling unprecedented complex-scene accuracy with linear complexity while optimizing semantic alignment, structural consistency, and textural richness.

## 7    ETHICS STATEMENT

This work has been conducted in accordance with the ICLR Code of Ethics. Our research does not involve human subjects, sensitive personal information, or personally identifiable data. All datasets employed in this study are publicly available, well-documented, and widely adopted in the research community, ensuring both accessibility and reproducibility. In preparing our experiments, we have carefully reviewed the terms of use and licensing conditions of these datasets, and we have taken steps to ensure that intellectual property rights and community norms are respected.

We are aware of potential societal impacts that may arise from advances in AI research, including challenges related to fairness, bias, security, and possible misuse of generative models. While our method is primarily designed to improve technical performance and enable deeper understanding of multimodal learning, we stress that such technologies should always be applied responsibly, with attention to ethical, social, and legal boundaries.

To promote transparency and reproducibility, we have made deliberate efforts to clearly document our training configurations, evaluation protocols, and experimental setups. We believe that such practices are essential not only for advancing scientific rigor but also for fostering responsible innovation in artificial intelligence.

## 8    REPRODUCIBILITY STATEMENT

To ensure the reproducibility of our work, we have taken a series of careful steps in both methodology and documentation. First, we provide a anonymized implementation of our system, which can be accessed at the following link: [https://anonymous.4open.science/r/Discan-F7FB/].

Our study employs retrieval-augmented generation (RAG) techniques, where retrieval is performed using CLIP similarity as the retrieval metric. Specifically, we compute cosine similarity between input queries and candidate features, and we select the Top-1 retrieved image or text as the corresponding reference. This ensures that the retrieval process is robust and can handle entirely unseen image–text pairs from the dataset. The retrieved features are then integrated into the model via a retrieval input module, which facilitates downstream reasoning and improves performance on challenging scenarios.

For image feature representation, we leverage the DC-AE model, specifically for "mit-han-lab/dc-ae-f32c32-sana-1.1". This model serves as a latent-space feature compressor, which effectively reduces the dimensionality of image representations while preserving semantic content. Additionally, we adopt random cropping and rotation augmentations during training to mitigate overfitting, ensuring that our model generalizes beyond specific patterns in the training data.

We also report all datasets used in our experiments, which are publicly available and widely adopted in the community. To facilitate replication, we provide a complete description of dataset preprocessing steps, including image normalization procedures, and the precise filtering rules applied. Furthermore, hyperparameter settings, including learning rate schedules, batch sizes, optimizer choices, and random seeds, are fully disclosed. Multiple experimental runs were conducted to verify the stability of results.

To enhance transparency, we detail the computational infrastructure used in our experiments, including GPU/CPU models, memory specifications, and operating systems. This ensures that reviewers and future researchers can align their environments with ours. We further include ablation studies to examine the contribution of different components, thereby enabling a more fine-grained understanding of how each design choice influences performance.

In summary, reproducibility has been a central priority of our research. We provide (i) anonymized source code, (ii) documentation of datasets and preprocessing steps, (iii) hyperparameter settings and random seeds. Together, these efforts in line with the ICLR guidelines on research transparency and reproducibility.

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

# Appendix

## The Use of Large Language Models (LLMs)

In accordance with the ICLR policy on the use of large language models, we confirm that no large language models were employed in the ideation, research, or writing of this work. Any text, code, or experimental results presented in this paper were entirely generated and validated by the authors without reliance on LLMs.

## Computational Environment

We summarize the hardware and software environment used in our experiments in Table 1.

| Component | Specification |
|---|---|
| GPU Model | NVIDIA RTX 4090 (24GB) |
| CPU Model | Intel Core i7-14700 (28 cores) |
| Memory | 32GB |
| Operating System | Ubuntu 22.04.5 LTS |

Table 1: Hardware and software setup used.

## Training Configurations

The training configurations for MS-COCO and MM-CelebA are summarized in Table 2.

| | MS-COCO 256 | MultiModal-CelebA |
|---|---|---|
| Autoencoder $f$ | 32 | 32 |
| $z$-shape | $32 \times 8 \times 8$ | $32 \times 8 \times 8$ |
| Patch size | 1 | 1 |
| Channels | 768 | 768 |
| Depth | 24 | 24 |
| Optimizer | AdamW | AdamW |
| Batch size | 256 | 64 |
| Learning rate | 1e-4 | 1e-4 |
| Weight decay | 0 | 0 |
| EMA rate | 0.9999 | 0.9999 |
| Warmup steps | 0 | 0 |

Table 2: Configurations for MS-COCO and MM-CelebA.

## Cross-Mamba computation

We formalize the computational framework of Cross-Mamba, which enables synergistic fusion of spatial and spectral features through dual-path state-space modeling. Let $\mathbf{F}_{\mathrm{sp}} \in \mathbb{R}^{L_s \times d}$ and $\mathbf{F}_{\mathrm{freq}} \in \mathbb{R}^{L_f \times d}$ denote spatial and spectral feature sequences, respectively. The dual-path SSM transformations are governed by:

$$\text{Path A:} \mathbf{Y}_a = \text{SSM}_{\Theta_a}(\mathbf{X}_{\mathrm{freq}}), \Theta_a = \{\mathbf{A}, \Delta, \mathbf{B}_{\mathrm{freq}}, \mathbf{C}_{\mathrm{sp}}\},$$
$$\text{Path B:} \mathbf{Y}_b = \text{SSM}_{\Theta_b}(\mathbf{X}_{\mathrm{sp}}), \Theta_b = \{\mathbf{A}, \Delta, \mathbf{B}_{\mathrm{sp}}, \mathbf{C}_{\mathrm{freq}}\}. \quad (1)$$

where $\mathbf{A} \in \mathbb{R}^{N \times N}$ and $\Delta \in \mathbb{R}$ are shared parameters maintaining consistent state dynamics. The cross-modal projections are defined as:

$$\mathbf{C}_{\mathrm{sp}} = \text{Linear}_c(\mathbf{F}_{\mathrm{sp}}), \mathbf{B}_{\mathrm{freq}}, \mathbf{X}_{\mathrm{freq}} = \text{Linear}_{bx}(\mathbf{F}_{\mathrm{freq}}),$$
$$\mathbf{C}_{\mathrm{freq}} = \text{Linear}_c(\mathbf{F}_{\mathrm{freq}}), \mathbf{B}_{\mathrm{sp}}, \mathbf{X}_{\mathrm{sp}} = \text{Linear}_{bx}(\mathbf{F}_{\mathrm{sp}}). \quad (2)$$

Discretizing the continuous SSM via zero-order hold yields:

$$\bar{\mathbf{A}} = \exp(\Delta\mathbf{A}), \bar{\mathbf{B}}_{\mathrm{freq}} = (\Delta\mathbf{A})^{-1}(e^{\Delta\mathbf{A}} - \mathbf{I})\mathbf{B}_{\mathrm{freq}},$$
$$\bar{\mathbf{B}}_{\mathrm{sp}} = (\Delta\mathbf{A})^{-1}(e^{\Delta\mathbf{A}} - \mathbf{I})\mathbf{B}_{\mathrm{sp}}. \quad (3)$$

The recurrent forms for time step $k$ are:

$$h_k^{(a)} = \bar{\mathbf{A}}h_{k-1}^{(a)} + \bar{\mathbf{B}}_{\mathrm{freq}}x_k^{(\mathrm{freq})}, y_k^{(a)} = \mathbf{C}_{\mathrm{sp}}h_k^{(a)},$$
$$h_k^{(b)} = \bar{\mathbf{A}}h_{k-1}^{(b)} + \bar{\mathbf{B}}_{\mathrm{sp}}x_k^{(\mathrm{sp})}, y_k^{(b)} = \mathbf{C}_{\mathrm{freq}}h_k^{(b)}, \quad (4)$$

Expanding the recurrence reveals the attention-like formulation:

$$\mathbf{Y}_a = \mathbf{C}_{\mathrm{sp}} \sum_{j=1}^{L_f} \left( \prod_{k=j+1}^{L_f} \bar{\mathbf{A}}_k \right) \bar{\mathbf{B}}_{\mathrm{freq},j} \mathbf{X}_{\mathrm{freq},j},$$
$$\mathbf{Y}_b = \mathbf{C}_{\mathrm{freq}} \sum_{j=1}^{L_s} \left( \prod_{k=j+1}^{L_s} \bar{\mathbf{A}}_k \right) \bar{\mathbf{B}}_{\mathrm{sp},j} \mathbf{X}_{\mathrm{sp},j}. \quad (5)$$

Applying the exponential property $\prod \exp(\mathbf{M}_k) = \exp(\sum \mathbf{M}_k)$:

$$\mathbf{Y}_a = \mathbf{C}_{\mathrm{sp}} \sum_{j=1}^{L_f} \exp \left( \sum_{k=j+1}^{L_f} \Delta_k \mathbf{A} \right) \bar{\mathbf{B}}_{\mathrm{freq},j} \mathbf{X}_{\mathrm{freq},j},$$
$$\mathbf{Y}_b = \mathbf{C}_{\mathrm{freq}} \sum_{j=1}^{L_s} \exp \left( \sum_{k=j+1}^{L_s} \Delta_k \mathbf{A} \right) \bar{\mathbf{B}}_{\mathrm{sp},j} \mathbf{X}_{\mathrm{sp},j}. \quad (6)$$

The final fusion integrates both paths through linear projection:

$$\mathbf{Y} = \text{Linear}_{\mathrm{fuse}}\left([\mathbf{Y}_a; \mathbf{Y}_b]\right), = \mathbf{W}_f [\mathbf{Y}_a \| \mathbf{Y}_b] + \mathbf{b}_f. \quad (7)$$

This establishes bidirectional co-modulation: Spatial features $\mathbf{F}_{\mathrm{sp}}$ guide spectral synthesis via $\mathbf{C}_{\mathrm{sp}}$ in Path A, while spectral features $\mathbf{F}_{\mathrm{freq}}$ regularize spatial generation via $\mathbf{C}_{\mathrm{freq}}$ in Path B. The shared state transition matrix $\mathbf{A}$ ensures coherent integration of complementary representations while maintaining linear complexity $\mathcal{O}(L)$.

## Structure-Aware Transfer Computation

To better understand how our Cross-Mamba module propagates structural priors from text to visual features, we formulate its computation as a structure-aware transformation process. This builds on a state-space formulation adapted from (Ali, Zimerman, and Wolf 2025), where SSM dynamics simulate attention-like interactions via parameterized recurrence.

We start from the standard SSM update:

$$\mathbf{h}_k = \bar{\mathbf{A}}_k\mathbf{h}_{k-1} + \bar{\mathbf{B}}_k\mathbf{x}_k, \mathbf{y}_k = \mathbf{C}_k\mathbf{h}_k. \quad (8)$$

To integrate structure from the semantic anchor $\mathbf{A}\_\text{prim}$, we map the intermediate parameters as:

$$\mathbf{C}_i = W_C(\mathbf{x}_i), \Delta_k = \text{ReLU}(W_\Delta(\mathbf{x}_k)),$$
$$\bar{\mathbf{A}}_k = \exp(\Delta_k \cdot \mathbf{A}), \bar{\mathbf{B}}_j = \Delta_j \cdot W_B(\mathbf{x}_j). \quad (9)$$

We then rewrite the output at timestep $i$ as the accumulated contribution from all previous positions:

$$\mathbf{y}_i = \sum_{j=1}^{i} \mathbf{C}_i \left( \prod_{k=j+1}^{i} \bar{\mathbf{A}}_k \right) \bar{\mathbf{B}}_j \mathbf{x}_j, \qquad (10)$$

This can be interpreted as directional modulation, where the structural role of each $\mathbf{x}_j$ is gated by $\bar{\mathbf{B}}_j$ and modulated by state recurrence $\bar{\mathbf{A}}_k$. The final update mimics cross-attention:

$$\mathbf{y}_i = \mathbf{Q}_i \cdot \mathbf{H}_{i,j} \cdot \mathbf{K}_j \cdot \mathbf{x}_j, \qquad (11)$$

where:

$$\mathbf{Q}_i = W_C(\mathbf{x}_i), \mathbf{K}_j = \mathrm{ReLU}(W_\Delta(\mathbf{x}_j) \cdot W_B(\mathbf{x}_j)),$$

$$\mathbf{H}_{i,j} = \exp\left( \sum_{k=j+1}^{i} W_\Delta(\mathbf{x}_k) \cdot \mathbf{A} \right). \qquad (12)$$

This shows that the output $\mathbf{y}_i$ can be seen as modulated retrieval over prior positions, where structural priors are encoded in $\mathbf{K}_j$, and state propagation is captured by $\mathbf{H}_{i,j}$. Importantly, $\mathbf{Q}_i$ determines how content (i.e., current spatial features) interacts with these priors.

By making $\mathbf{B}, \Delta$ dependent on structure ($\mathbf{x}_j = \mathbf{F}_{\mathrm{down}}$) and $\mathbf{C}$ on content ($\mathbf{x}_i = \mathbf{E}_v$), the SSM transition simulates a structure-aware transformation that aligns visual layout with semantic references. This explains how our Cross-Mamba module transfers reference-aligned priors while preserving directional modulation and spatial coherence.

## SP-DSS Fusion modal code

The SP-DSS Fusion module integrates textual and visual information through a multi-directional state-space framework, as outlined in Algorithm 1. It first concatenates the two feature streams and applies a shared linear layer to derive global dynamic parameters $(\mathbf{A}, \Delta)$ that summarize common temporal or spatial dependencies across modalities. The model then launches four directional scans; in each scan $i$, separate projections create modality-specific parameters $(\mathbf{B}^{(i)}, \mathbf{C}^{(i)}, \mathbf{X}^{(i)})$, which are fed into a structured state-space model $\mathrm{SSM}\_\theta\_i$. These directional outputs are spatially restored via $\mathcal{R}\_i^{-1}$ and summed, ensuring that both local and long-range cues are captured from every orientation.

## DC-AE Inference Pipeline

We use a pre-trained DC-AE (Diffusion-Compatible AutoEncoder) model from the Diffusers library to compress and reconstruct images in Algorithm 2. The input image is first normalized and converted to a tensor. It is then encoded into a latent representation using the encoder module of DC-AE. The latent vector is decoded back into image space by the decoder.

## More qualitative examples

We present our generated samples of MS-COCO in Figure 1, MM-CelebA-HQ-256 in Figure 2, Cifar-10 in Figure 3.

---

**Algorithm 1: SP-DSS Fusion with Quadrangular Scanning**

**Input**: Text features $\mathbf{Z}_t$, Visual features $\mathbf{Z}_v$
**Parameter**: Specific affine parameters $\gamma_t, \beta_t, \gamma_v, \beta_v$
**Output**: Enhanced features $\mathbf{O}'_t, \mathbf{O}'_v$

1: $\mathbf{Z}_{\mathrm{fused}} \leftarrow \mathrm{Concat}[\mathbf{Z}_t; \mathbf{Z}_v]$ {Sequence length $L = L_t + L_v$}
2: $\mathbf{A}, \Delta \leftarrow \mathrm{Linear}_{\mathrm{shared}}(\mathbf{Z}_{\mathrm{fused}})$ {Shared dynamics parameters}
3: $\mathbf{Y} \leftarrow \mathbf{0}$ {Initialize output accumulator}
4: **for** $i = 1$ **to** $4$ **do**
5: $\quad \mathbf{B}_t^{(i)}, \mathbf{C}_t^{(i)}, \mathbf{X}_t^{(i)} \leftarrow \mathrm{Linear}_{\mathrm{dir}\_i}(\mathbf{Z}_t)$
6: $\quad \mathbf{B}_v^{(i)}, \mathbf{C}_v^{(i)}, \mathbf{X}_v^{(i)} \leftarrow \mathrm{Linear}_{\mathrm{dir}\_i}(\mathbf{Z}_v)$
7: $\quad \mathbf{B}^{(i)} \leftarrow \mathrm{Concat}[\mathbf{B}_t^{(i)}, \mathbf{B}_v^{(i)}]$
8: $\quad \mathbf{C}^{(i)} \leftarrow \mathrm{Concat}[\mathbf{C}_t^{(i)}, \mathbf{C}_v^{(i)}]$
9: $\quad \mathbf{X}^{(i)} \leftarrow \mathrm{Concat}[\mathbf{X}_t^{(i)}, \mathbf{X}_v^{(i)}]$
10: $\quad \mathbf{Y}^{(i)} \leftarrow \mathrm{SSM}_{\theta_i}(\mathbf{X}^{(i)})$ {$\theta_i = \{\mathbf{A}, \Delta, \mathbf{B}^{(i)}, \mathbf{C}^{(i)}\}$}
11: $\quad \mathbf{Y} \leftarrow \mathbf{Y} + \mathcal{R}_i^{-1}(\mathbf{Y}^{(i)})$ {Spatial restoration and aggregation}
12: **end for**
13: $\mathbf{Y} \leftarrow \mathrm{LayerNorm}(\mathbf{Y})$
14: $\mathbf{Y} \leftarrow \mathrm{Linear}_{\mathrm{proj}}(\mathbf{Y})$
15: $\mathbf{O}_t, \mathbf{O}_v \leftarrow \mathrm{Split}(\mathbf{Y}, [L_t, L_v])$
16: $\mathbf{O}'_t \leftarrow \sigma(\gamma_t) \odot \mathbf{O}_t + \beta_t$ {Text modulation}
17: $\mathbf{O}'_v \leftarrow \sigma(\gamma_v) \odot \mathbf{O}_v + \beta_v$ {Visual modulation}
18: **return** $\mathbf{O}'_t, \mathbf{O}'_v$

---

**Algorithm 2: Image Processing via DC-AE**

**Input**: Image $\mathbf{I}$
**Model**: Pretrained Autoencoder `DC-AE`
**Output**: Reconstructed image $\hat{\mathbf{I}}$

1: `DC-AE` $\leftarrow$ `LoadModel`("XXX")
2: $\mathbf{I}_{\mathrm{tensor}} \leftarrow$ `Normalize`(`ToTensor`($\mathbf{I}$), $\mu=0.5, \sigma=0.5$)
3: $\mathbf{I}_{\mathrm{tensor}} \leftarrow \mathbf{I}_{\mathrm{tensor}}[None]$ {Add batch dimension}
4: $\mathbf{I}_{\mathrm{tensor}} \leftarrow \mathbf{I}_{\mathrm{tensor}}.to(\mathrm{cuda})$
5: $\mathbf{z} \leftarrow$ `DC-AE.encode`($\mathbf{I}_{\mathrm{tensor}}$)`.latent` {Encode image to latent}
6: $\hat{\mathbf{I}}_{\mathrm{tensor}} \leftarrow$ `DC-AE.decode`($\mathbf{z}$)`.sample` {Decode latent to reconstructed image}
7: $\hat{\mathbf{I}}_{\mathrm{tensor}} \leftarrow \hat{\mathbf{I}}_{\mathrm{tensor}} \times 0.5 + 0.5$ {Denormalize to [0,1]}
8: `SaveImage`($\hat{\mathbf{I}}_{\mathrm{tensor}}$, "demo_dc_ae.png")
9: **return** $\hat{\mathbf{I}}$

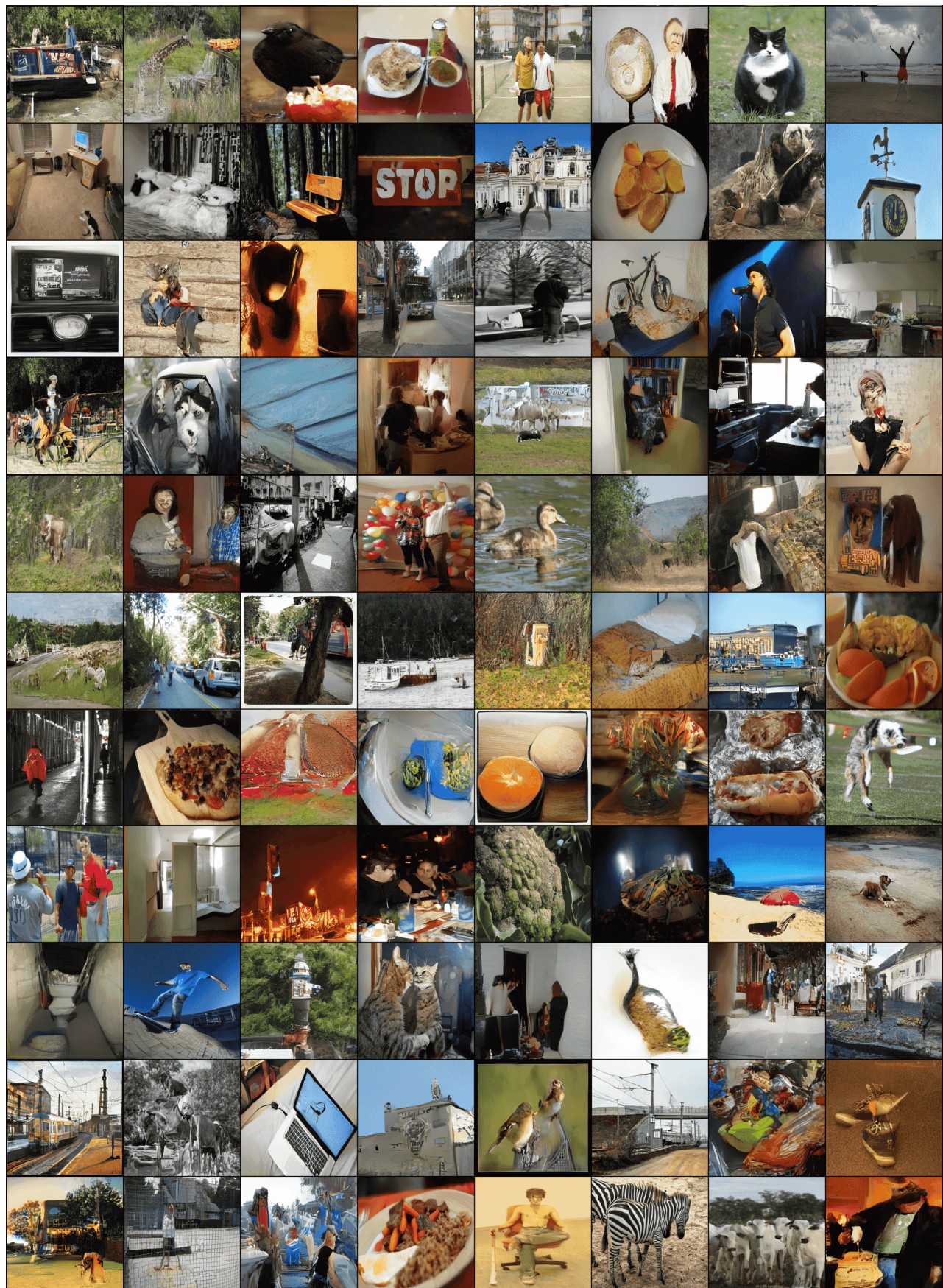

Figure 1: Uncurated generated samples of MS-COCO.

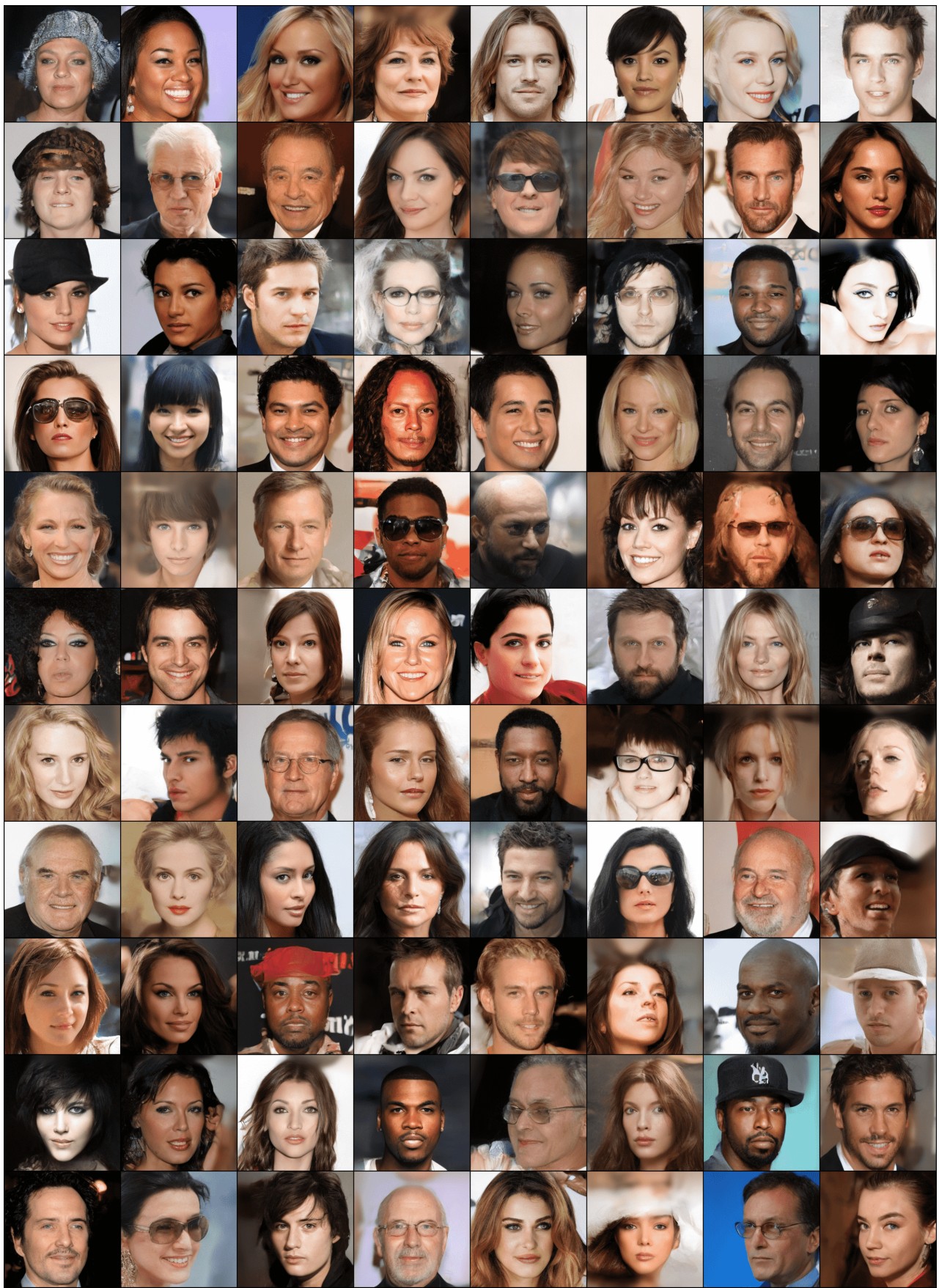

Figure 2: Uncurated generated samples of MM-CelebA-HQ-256.

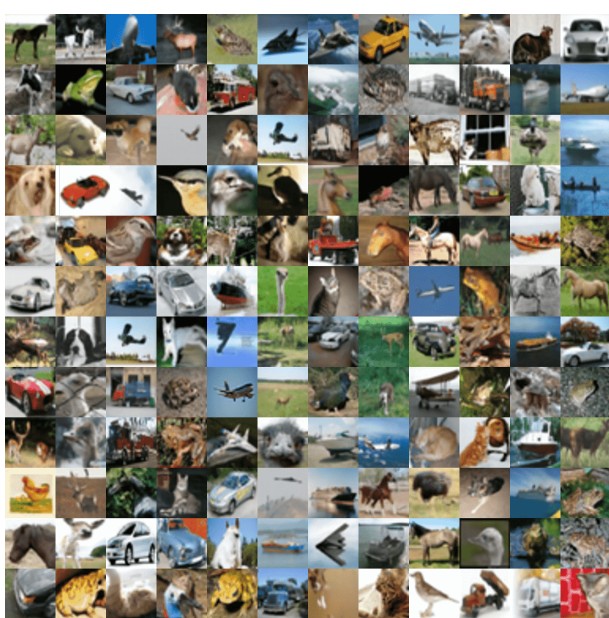

Figure 3: Uncurated generated samples of CIFAR-10.