# OpenReview forum: "DiScan: a quad-directional SSM diffusion framework"
_ICLR.cc/2026/Conference — ICLR 2026 Conference Withdrawn Submission_

### Official Review · Reviewer_bhV9 · 2025-10-18

**Soundness:** 3
**Presentation:** 3
**Contribution:** 2
**Rating:** 4
**Confidence:** 4

**Summary:**

The paper builds a text-to-image diffusion backbone using quad-directional SSM scanning with shared dynamics and direction-specific projections, plus a dual-phase retrieval module and a spectral–spatial fusion (wavelet + scanning). It reports FID gains over SSM baselines (e.g., +9.95 on COCO).

**Strengths:**

1. Clean decomposition into SP-DSS, DP-RAG, and USS-CP; readable figures.
2. Consistent improvements over the chosen SSM baselines.

**Weaknesses:**

1. Limited novelty; mostly architectural mixing.
2. Missing ablations isolating each part (directions, retrieval, spectral block) across multiple datasets.
3. Efficiency story is incomplete (throughput, memory) compared to DiT or other SSMs.

**Questions:**

1. What is the effect size of each component when added alone?
2. How is retrieval built (index size, features, latency)?
3. Any comparison to transformer diffusion (DiT) at matched FLOPs?

---

### Official Review · Reviewer_xdou · 2025-10-31

**Soundness:** 3
**Presentation:** 3
**Contribution:** 3
**Rating:** 4
**Confidence:** 2

**Summary:**

This paper introduces DiScan, a T2I diffusion framework based on State Space Models. It aims to fix common issues in previous methods, including texture blurring, object distortion, and poor text alignment. The authors propose three main components: a Shared-Projection Directional State-Space (SP-DSS) module, which is a quad-directional SSM that jointly scans text and visual features to improve spatial consistency; a Dual-Phase Retrieval-Augmented (DP-RAG) module, which retrieves reference images to inject semantic and structural guidance; a Unified Spectral-Spatial Co-Processing (USS-CP) block, which uses wavelet decomposition to capture high-frequency texture details.

**Strengths:**

1. The problem that this paper tackles is important. Improving efficiency of generative models by linear attention is fundamental and has great applications.
2. The architecture is well-motivated. Each of the three components is designed to solve a specific failure mode of generative models. The SP-DSS addresses the spatial continuity issues of simple row-scanning SSMs, the DP-RAG targets structural and alignment failures, and the USS-CP tackles texture blurring.
3. This paper is well presented, with clear figure illustrating the pipeline and quantitative results showing the ablation effects.

**Weaknesses:**

1. The paper's core motivation for using SSMs is efficiency. However, the DP-RAG module introduces a retrieval step (Top-1 CLIP score search) at inference time. This retrieval process has its own significant computational cost. The authors should report metrics about efficiency, such as latency, FLOPs, etc.
2. This paper makes claims about a "scalable paradigm", but experiments are up to 256x256 resolution. The benefits of linear attention over quadratic attention should be magnified for high-resolution image synthesis. This undermines the paper's claim of scaling.

*Reviewer's Acknowledgement:* I am not familiar with SSMs. I do not know why this paper is assigned to me. I appreciate that the authors would address my concerns or point out any misunderstanding.

**Questions:**

1. From Table 1 and 2, the best performing variant is DiScan_{t2t}. I can see DiScan_{t2i} is the one shown in ablation studies of Table 5. But I failed to locate where DiScan_{t2t} is explained.

---

### Official Review · Reviewer_ZHq1 · 2025-11-01

**Soundness:** 1
**Presentation:** 1
**Contribution:** 2
**Rating:** 2
**Confidence:** 4

**Summary:**

This paper tries to improve the state space model-based text-to-image diffusion model. It proposes three new modules: (1) SP-DSS for cross-modal alignment, (2) DP-RAG, retrieval-based module that leverage reference feature to improve generation, and (3) USS-CP that combines wavelet decomposition with bidirectional scanning.

**Strengths:**

1. The design of three proposed modules are introduced clearly.

**Weaknesses:**

1. The proposed DP-RAG module relies additional reference images, which makes the model not a traditional text-to-image setting (with only text input). This may bring unfair advantage. Can baseline models use the reference images as well?
2. One advantage of the proposed modules is its parameter efficiency. However, compared with baseline models, it introduces three additional modules, which includes more trainable parameters. Including a comparisons of numbers of parameters help understand the trade-off between performance gain and training/saving costs.
3. Similarly, the paper should report inference time and compared with baseline models.
4. In experiments, why the baseline models are not always included in all datasets? For example, Zigzag-2 does not occur in Table 2, 3, VisionMamba does not occur in Table 1,3, and Table 4 only includes Zigma and MM-DiT.
5. In Table 1 and 2, what is the difference between the bold **$DiScan_{t2i}$** and normal $DiScan_{t2i}$? In Table 3, what is “Ours”? Is it $DiScan_{t2i}$?
6. Paper presentation and formats. The paper seems to have been completed in a rush, and there are many formatting inconsistencies, such as a repeated Section 5.4 title, inconsistency in whether subsections include a period (e.g., Section 4.5), different font sizes (e.g., Figures 4 and 6), varying distances between captions and tables (e.g., Tables 1, 2, and 3), and inconsistent titles between the OpenReview page and the submitted PDF.
7. (Personal comment) The contribution could potentially be limited. The diffusion transformer architecture has achieved significant success, and compared with it, the SSM architecture does not show a clear advantage at this stage. Even if it is theoretically more efficient because it avoids quadratic computation, this efficiency should be demonstrated on large images—which is not the case, as the current experiments are conducted on relatively small images. Given this, the contribution of this work could be limited, since it aims to improve an architecture that is not widely adopted and has not been proven superior to the DiT architecture.

**Questions:**

Please see the weakness part.

---

### Official Review · Reviewer_ty14 · 2025-11-04

**Soundness:** 3
**Presentation:** 2
**Contribution:** 2
**Rating:** 4
**Confidence:** 4

**Summary:**

This paper proposes DiScan, a novel text-to-image diffusion framework designed to fix common issues like texture blurring, shape distortion, and poor text alignment. Its architecture combines three key innovations: a quad-directional SSM for better spatial and semantic coherence , a dual-stage retrieval module (DP-RAG) that uses reference images to preserve object structure , and a spatial-frequency fusion block (USS-CP) that uses wavelets to capture fine textures and details. The authors demonstrate that DiScan significantly outperforms existing SSM-based models, achieving substantial FID score improvements on COCO, CelebA-HQ, and CIFAR-10.

**Strengths:**

1. The authors introduced a mamba-based text-to-image Diffusion model, which aims at breaking the high memory bottle-neck of Unet, and the quadratic complexity with respect to the sequence length of ViT.

2. The authors developed three strategies to improve the vanilla SSM-based DM. The first strategy is to combine four sweeping scan directions together with parameter sharing, which may improve the continuity of the spatial information of the image. The second strategy is to incorporate the cross-attention between image and text. The third strategy is to separate the high frequency and low frequency components. Though these three strategies are somehow not novel, it does improve the generation quality.

**Weaknesses:**

1. In the Introduction section, the authors claim that "due to the simple cross-attention mechanisms". I don't think this is the main reason for the alignment issue for the text and image content. In addition, this paper also adopts the "simple cross-attention mechanisms" without any improvements on cross-attention.

2. On line 54, there is a typo "This resuts".

3. This paper combined four sweeping scans in one framework. However, there is no any ablation study on if these simple combination has improved the performnace.

4. In the related works, there are many text-to-imgae model based on SSM, which is closely related to this paper. Therefore, it is necessary to add on paragraph in related work section.

5. Many notations are not standard and rigorous. For example, is W_t(I) a function of I or the product between W_t and I? If it is the former one, it should be W_t(I) and if it is the later one, it should be W_t * I or W_t I. From the paper, in my understanding, I think it should be the latter one.

6. The reasons for sharing the projection matrix A_i and Delta_i seem tricky; there are no motivations and reasons for sharing these two matrices among all scaning direction.

7. The authors adopts four sweeping scan directions in this paper, thus there are four B_v, C_v, X_v. However, from Eq. (6), the authors showed that there are also four B_t, C_t and X_t. In this paper, how to scan the one-dimension text data in four directions?

8. In DP-RAG module, the authors adopts the cross-attention block, which scales as O(N^2) with length N. Therefore, the advantages of the linear complexity of the SSM model do not exist.

**Questions:**

See the weakness section above.

---

### Note · Authors · 2025-11-15

I have read and agree with the venue's withdrawal policy on behalf of myself and my co-authors.